# Trace Organic Compound Removal from Wastewater Reverse-Osmosis Concentrate by Advanced Oxidation Processes with UV/O_3_/H_2_O_2_

**DOI:** 10.3390/ma13122785

**Published:** 2020-06-19

**Authors:** Aviv Kaplan, Hadas Mamane, Yaal Lester, Dror Avisar

**Affiliations:** 1The Water Research Center, Porter School for Environment and Earth Sciences, Faculty of Exact Sciences, Tel Aviv University, Tel Aviv 69978, Israel; avivkaplan@tauex.tau.ac.il; 2School of Mechanical Engineering, Faculty of Engineering, Tel Aviv University, Tel Aviv 69978, Israel; hadasmg@tauex.tau.ac.il; 3Environmental Technologies, Department of Advanced Materials, Azrieli College of Engineering, Jerusalem 9103501, Israel; lester.yaal@gmail.com

**Keywords:** advanced oxidation, wastewater, reverse osmosis, concentrate, trace organic compound, micropollutant

## Abstract

Advanced technologies, such as reverse osmosis (RO), allow the reuse of treated wastewater for direct or indirect potable use. However, even highly efficient RO systems produce ~10–15% highly contaminated concentrate as a byproduct. This wastewater RO concentrate (WWROC) is very rich in metal ions, nutrients, and hard-to-degrade trace organic compounds (TOrCs), such as pharmaceuticals, plasticizers, flame retardants, and detergents, which must be treated before disposal. WWROC could be up to 10 times more concentrated than secondary effluent. We examined the efficiency of several advanced oxidation processes (AOPs) on TOrC removal from a two-stage WWROC matrix in a pilot wastewater-treatment facility. WWROC ozonation or UV irradiation, with H_2_O_2_ addition, demonstrated efficient removal of TOrCs, varying between 21% and over 99% degradation, and indicating that radical oxidation (by HO·) is the dominant mechanism. However, AOPs are not sufficient to fully treat the WWROC, and thus, additional procedures are required to decrease metal ion and nutrient concentrations. Further biological treatment post-AOP is also highly important, to eliminate the degradable organic molecules obtained from the AOP.

## 1. Introduction

More than 2 billion people live under high water stress, which is expected to increase due to effects of climate change and global warming [1]. With the population in cities growing rapidly, it is projected that by 2050, more than 66% of the human population will reside in urban areas [2]. Thus, the key issues to consider involve water and wastewater treatment, water availability, and land shortages. Water reuse is the best economic and environmental solution, and possibly the only viable one, for the increase in urban water consumption and depletion of water sources [3].

Consequently, there is a need for advanced treatment of wastewater that generates high-quality potable water. Today, desalination technologies such as reverse osmosis (RO) or nanofiltration are attractive for wastewater reclamation and reuse [4]. RO of wastewater (subsequent to biological treatment) can produce high-quality reclaimed water [5,6,7,8,9] by rejecting contaminants such as metal ions, nitrates, viruses, pharmaceuticals, and pesticide residues. Various combinations of bioreactors followed by RO processes demonstrate recovery of 80–90% of the water [10,11,12], while the remaining 10–20% is rejected as byproduct.

Wastewater RO concentrate (WWROC) is the main byproduct of RO treatment, which usually contains high concentrations of salt, organic micropollutants (refractory compounds), effluent organic matter, metals, nitrogen, and phosphates [13]. Since WWROC is five–nine times more concentrated than secondary effluent (depending on process efficacy), the concentration of pollutants is expected to increase accordingly. Toxic metals, for example, are present in secondary effluent at low concentrations of 1–100 µg/L [11]. These concentrations of these metals can rise to 100–1000 µg/L if the RO process is efficiently operated, increasing their toxic effects. High levels of nutrients, mainly nitrogen and phosphorus (as NO_3_^−^ and PO_4_^3−^) have been found in WWROC, ranging from 20–155 mg/L and 5–65 mg/L, respectively [14,15,16]. As a result, direct disposal of the WWROC will pose a threat to aquatic ecosystems. Various methods are being studied and applied by both researchers and the industry to reduce the toxic elements in the WWROC or its final volume [17]. The zero liquid discharge approach, combining thermal evaporators, crystallizers, and spray driers, is a good example, but requires high operation costs and land use [18]. Since the RO process is usually employed as a tertiary treatment, after the secondary biological stage, the dissolved organic matter is mostly non-biodegradable. Therefore, it contains high levels of refractory organic compounds and micropollutants, characterized by aromatic rings and double bonds, such as pharmaceutical residues, personal care products, plasticizers, endocrine disruptors, pesticides, herbicides, and more [19]. Some of those micropollutants (also known as trace organic compounds—TOrCs) have the potential to bioaccumulate in biota [20], and several of them are considered toxic to aquatic organisms [21]. TOrCs are present at low concentrations in the environment (usually in the ng/L–µg/L range in aquatic environments) and can be found in various aqueous environments, impacted by human activity and waste [22]. A typical example of TOrCs are pharmaceuticals, which often end up in water bodies via wastewater/effluent streams [23,24]. 

Alternative techniques have been studied in the last few decades for efficient TOrC removal, including soil aquifer treatment [25], UV photolysis, and advanced oxidation processes (AOPs) using UV/TiO_2_ [26,27,28], UV/H_2_O_2_ [29,30,31], and ozonation [32,33]. AOPs have been successfully implemented on WWROC matrices for TOrC removal [13,15,17,27]. The advantage of the AOP is that it is a destructive process, with a chemical–physical degradation mechanism that oxidizes molecules such that they become more biodegradable or may even reach full mineralization. 

However, AOP efficiency for TOrC removal is sensitive to the water matrix, which can vary among WWROCs produced in different facilities or even on different days in the same facility, depending on the influent parameters [9]. Nevertheless, the implementation of UV- and ozone-based AOP treatments on WWROC to remove TOrCs has not been sufficiently studied. Our aim was to evaluate the removal of seven pharmaceuticals from real WWROC, generated by a pilot facility treating municipal wastewater using the AOPs: UV/H_2_O_2_, ozone (O_3_), and O_3_/H_2_O_2_.

## 2. Materials and Methods

### 2.1. Analyte Selection

Seven pharmaceuticals were selected based on their prevalence in treated wastewater in Israel [25] and their different oxidation-reaction rates with ozone and HO (K_O3_, K_OH_), as specified in Table 1.

### 2.2. WWROC Sample Preparation

The WWROC samples were collected from the Technion’s effluent-desalination pilot plant, located at the wastewater treatment plant (WWTP) of Nir Ezyon, Israel (Figure 1). The pilot plant was already described in detail elsewhere [15]. Generally, it comprised of an ultrafiltration (UF) system, followed by a two-stage RO unit, generate permeate at about 85% total recovery. Sulfuric acid was added after UF and antiscalant (Osmotech 1262, Kurita, Ludwigshafen, Germany) was added after the first RO stage to attain a pH of 6.0–6.5, in order to prevent precipitation of scaling salts on the membranes. Feed water were the secondary effluent produced by the Nir Ezyon WWTP. The WWROC chemical parameters are presented in Table 2.

All WWROC samples were filtered through a 2.7-µm filter (GF/D Whatman, Buckinghamshire, UK) before handling. Since samples were collected on different dates, they were adjusted to the desired pH, when necessary, and spiked with TOrCs (at final concentrations of ~20 µg/L for iohexol (IHX) and ~5 µg/L for all other TOrCs, Sigma-Aldrich, Rehovot, Israel) to ensure sufficient concentrations for reliable detection. Samples were adjusted to three different conditions to compare AOP efficiencies:Samples adjusted to pH 6,Samples adjusted to pH 6 and addition of 60 ppm H_2_O_2_–1 ppm H_2_O_2_ per 1 ppm initial dissolved organic carbon (DOC),Increase in pH to 10.5 with 1N NaOH and removal of the precipitate with a 2.7-µm filter.

Applying different AOPs on the different sample conditions are expected to result in significant degradation efficiency variations, due to different oxidation mechanisms of the TOrCs—direct reaction with the oxidative agent or indirect mechanism (reaction with OH· which are formed by the oxidative agent)—Table 3.

### 2.3. Laboratory-Scale AOP Experiments

#### 2.3.1. Ozone-Based AOP Experiments

The Laboratory ozone experiments were performed using a semi-continuous batch reactor which described elsewhere [37]. Briefly, pure oxygen is flowing at a rate of 1.5 L/min through O_3_ generator, which enrich the oxygen up to 4 g/h. An inlet gauge set the flow at 0.4 L/min and splits it to inlet ozone gas analyzer and to a glass diffuser inside a 0.5 L cylinder glass reactor, containing 0.25 L of WWROC sample. Ozone gas not reacting with the sample flows to an outlet gas analyzer. The accumulated transferred ozone dose (TOD) can be estimated as [37]:(1)TOD(mgL)=∑(Co3,in−Co3,out)mgL×gas flow rateLmin×ΔtminSample volume L
where, C_O3,in_ is the inlet O_3_ concentration, C_O3,out_ is the outlet O_3_ concentration, and the measurement intervals (${\Delta t}_{\min}$) are one minute.

#### 2.3.2. UV Photolysis and AOP-Based Experiments

UV radiation experiments were performed by bench-scale UV collimated beam apparatus. The UV lamp (0.45 kW MP polychromatic, Ace-Hanovia, 7830, Vineland, NJ, USA) was irradiated through a circular opening, perpendicular to the surface of the WWROC sample. The incident irradiance was measured with a spectroradiometer (USB4000, Ocean, Rochester, NY, USA) over the 220–350 nm range (mW/cm^2^) and was used to calculate the average UV fluence rate, taking into account the different water factors (spectral absorbance, reflectance factor, and Petri factor). Each sample was transferred into a suitable 100 mL glass beaker with a PTFE stirrer and placed on a stirring plate located beneath the lamp’s circular opening at a fixed distance. UV fluence (mJ/cm^2^) was calculated by multiplying the average fluence rate by the exposure time.

### 2.4. Analysis

#### 2.4.1. Sample Preparation

Samples before and after AOP treatment underwent a solid-phase extraction procedure for clean-up and concentration. A 500 mg/6 mL Oasis HLB cartridge (Waters, Milford, MA, USA) was mounted on a vacuum manifold, equilibrated with 10 mL methanol (ULC/MS, Biolab, Jerusalem, Israel) and 10 mL deionized water. A 10-mL volume of sample was loaded through the HLB cartridge, then washed with 2% methanol. The cartridge was vacuum-dried for 10 min, followed by elution of 6 mL of water: methanol: acetonitrile (ULC/MS, Biolab, Jerusalem, Israel) (90:5:5 *v/v*%) solution. This eluent was mixed with 30 mL methanol (E1). The cartridge was then vacuum-dried again and eluted with 8 mL acetonitrile (E2). Finally, the cartridge was eluted with 8 mL methanol (E3). The obtained eluents—E1, E2, and E3—were evaporated individually at 45 °C under low nitrogen flow to dryness, reconstituted with 1 mL of water: methanol: formic acid (84.9:15:0.1 *v/v*%), and taken for HPLC–MS analysis. The results from eluents E1, E2, and E3 were calculated together for each sample.

#### 2.4.2. Analytical Measurements

Chromatographic separation of TOrCs in the samples was performed by HPLC (1100, Agilent, Santa Clara, CA, USA) equipped with a Kinetex biphenyl 100 × 3.0 mm 2.6-µm analytical column (Phenomenex, Torrance, CA, USA) at 40 °C. Injection volume was 100 µL and the flow rate was 0.5 mL/min. The mobile phase contained 0.1% formic acid (Merck, Burlington, MA, USA) in water (Solution A) and 0.1% formic acid in methanol (Solution B). A gradient program was applied, starting from 0–1 min (hold at 90% solution A), 1–5 min (to 50%), 5–20 min (to 10%), 20–23 min (hold at 10%), 23–25 min (change back to 90%), and 25–33 min (hold at 90% for column equilibration).

Detection and quantification were performed using a high-resolution MS (Q-TOF premier Waters, Milford, MA, USA) via an Electrospray Ionization (ESI) interface in positive mode. Data acquisition and evaluation were performed using Waters chromatography MassLynx software (v4.1, Waters, Milford, MA, USA). TOrCs were identified according to their retention time and exact mass [M + H] (Table 4). Additional quality parameters were measured as described in Table 5.

## 3. Results and Discussion

### 3.1. UV and UV/H_2_O_2_


The efficacy of direct UV photolysis at pH 6 and 10.5, and UV with added 60 ppm H_2_O_2_ at pH 6, for two different UV fluence values, on degradation of the selected TOrCs is presented in Figure 2. The degradation percentage in the presence of H_2_O_2_ was favorable for all compounds (compared to UV alone) at pH 6. Sulfamethoxazole (SMX) and particularly diclofenac (DCF) were susceptible to photolysis, with a minor improvement when H_2_O_2_ was added. Carbamazepine (CBZ), bezafibrate (BZF), and venlafaxine (VLX) were only marginally degraded by direct UV photolysis, but their degradation increased significantly (by 15–30%) in the presence of H_2_O_2_. IHX and lamotrigine (LMG) also showed low- to moderate degradation by UV photolysis, but there was no substantial improvement with the addition of H_2_O_2_. Except for DCF, all compounds demonstrated better degradation at pH 6 than at pH 10.5.

The superior efficiency of UV/H_2_O_2_ vs. UV alone at the same pH has already been demonstrated for several types of TOrCs [29,38,39]. In our case, the indirect mechanism of radical degradation seems to be more effective than photodegradation for all of the studied TOrCs. CBZ, BZF, and VLX have higher K_OH_ values than SMX, IHX, and LMG (Table 1), and the relative improvement in degradation is therefore higher. Nevertheless, the overall degradation of the studied TOrCs, even at a higher UV fluence and with the addition of H_2_O_2_, was lower than 50%, except for DCF and SMX which are known to have a high reaction rate under direct photolysis [40].

Other studies on TOrC removal from WWROC by UV/H_2_O_2_ [13,40] have also demonstrated sufficient degradation efficiency, even with lower H_2_O_2_/DOC ratios. However, the deviations in WWROC parameters from individual production processes differ significantly between sites [17], in addition to the impact of influent parameters, season and the RO setup [9], affecting the UV/H_2_O_2_ efficiency (for example, Justo et al. [13] demonstrated over 70% degradation for all of their studied TOrCs from ratios of 0.54–0.72 ppm H_2_O_2_/total organic carbon—TOC). Another major factor regarding UV radiation is the variety of possible setups. Every setup will result in different degradation efficiency (e.g., UV lamp locate above the sample surface, as on this study, versus UV lamp immersed inside the sample [26]). Therefore, an accurate comparison between treatments studied on WWROC from different sites is difficult to achieve. However, in comparison to the one-stage RO process in Justo et al. [13], this study was done with a two-stage RO process, characterized by higher values of DOC, conductivity, UVA_254_ and metal ions, indicating the presence of high concentrations of organic and inorganic compounds that may interfere with the degradation process (scavenging rate).

Even though a relatively high ratio of 1 ppm H_2_O_2_/1 ppm DOC was applied, H_2_O_2_ also has HO· scavenging characteristics [41]. At high H_2_O_2_ concentration, a competitive reaction between H_2_O_2_ and HO· becomes significant [31]:H_2_O_2_ + HO·→ HO_2_·+ H_2_O(2)

Therefore, the possibility that the H_2_O_2_ concentration used here (60 ppm) increased the scavenging rate over TOrCs degradation cannot be ruled out. In addition, the UV light absorbance of H_2_O_2_ at high concentrations may screen the TOrCs and reduce the rate of their direct photolysis [31].

### 3.2. OH· Scavengers

Except for H_2_O_2_, some other compounds found in wastewater are generally known as OH· scavengers. Those could be organic and inorganic molecules that also react with the OH·, since it is not a selective radical which react strongly with almost any compound. Such known scavengers are the unspecific effluents organic matter (EfOM) [40,42], usually expressed by TOC or DOC, inorganic anions as halides [40,43], carbonates and bicarbonates [44], and suspended particles [45]. Based on the work of Rosenfeldt and Linden [41], Sharpless and Linden [44], and Lester et al. [31] it was possible to calculate the OH· scavenging rate by the matrix (calculations not shown, see Appendix A). Briefly, para-chlorobenzoic acid (pCBA) was added into the WWROC matrix with different concentrations of H_2_O_2_ and UV radiation. The p-CBA degradation rate was compared to the total OH· formation by each concentration to calculate the scavenging rate. The obtained value of scavenging rate was 1.29 × 10^4^ s^−1^. This value is relatively low, similar to the values found in lakes [46] rather than wastewater, which is at least one-fold higher [46,47].

Low scavenging rate obtained here might be a result in the acidification stage before RO filtration, which eliminate carbonates/bicarbonates due to the relatively low pH (pH~6 or lower transform bicarbonates to carbonic acid). In addition, the Ultra-filtration (UF) step before the RO process (Figure 1) significantly reduces suspended particles, which also contributes to OH· scavenging. Finally, the quality of the effluent and its source (e.g., municipal wastewater from rural settlements and small towns), efficiency of the secondary biological treatment and seasonality can also have a major effect on the scavenging rate.

Overall, it seems that no major OH· scavenging is arisen by the studied WWROC matrix. However, this might be a result of the specific WWROC production process used here and cannot be deduced for every WWROC.

### 3.3. Ozone and O_3_/H_2_O_2_

The efficacy of direct ozonation at pH 6 and 10.5, and ozone with added 60 ppm H_2_O_2_ at pH 6, for two different ozonation times, for degradation of the selected TOrCs, is presented in Figure 3. Samples undergoing shorter ozonation time (5 min) demonstrated a lower degree of degradation for the slow and moderately reactive compounds. At the longer ozonation time (12 min), all of the fast and moderate reaction rate compounds demonstrated over 90% degradation for all tested conditions. The slow-reacting IHX and LMG also demonstrated much better degradation efficiency for all tested conditions for this time interval (a longer time interval results in a higher ozone dose). 

Comparing operational conditions, the high-pH sample demonstrated the highest degradation efficiency, 70% for both IHX and LMG (under the longer ozonation time). Encouraging, however, was that the sample at pH 6 with the addition of H_2_O_2_ (1 ppm for every ppm DOC) demonstrated good TOrC degradation results, only 10% lower than the high-pH condition for LMG and IHX. Compared to samples at pH 6 without H_2_O_2_, the degradation efficiency of the former was 15–24% higher.

Ozone reactions can degrade TOrCs via two pathways: direct (O_3_) and indirect (HO·) [48]. While the direct mechanism is specific for each compound and depends on its reaction rate (K_O3_), the indirect mechanism is relatively non-selective and usually much more reactive, since it involves radical reactions with the HO· generated during ozonation [49]. The reaction rate of a TOrC with HO· (K_OH_), as already discussed, is usually several orders of magnitude higher than that compound’s K_O3_ (Table 1).

In this study, four out of the seven compounds had relatively high K_O3_ values (CBZ, VLX, DCF, and SMX), hence their degradation by ozone was very rapid and did not allow distinguishing the effects of the different tested conditions. Although VLX is considered a fast ozone reactant, its K_O3_ is at the lower limit of this definition (Table 1) and at the lower ozone dose tested, it did not fully degrade. Hence VLX, BZF, LMG, and IHX are the compounds that should be examined for degradation efficacy under the different conditions, with a better distinction between compounds observed at the lower ozone dose interval (5 min of ozonation).

As implied by the results, there is an increment of 6–14% degradation (for each individual TOrC) by the addition of H_2_O_2_ to the WWROC ozonation at the 5-min interval, and 15–24% at the 12-min interval, when the pH is held constant at 6. The addition of H_2_O_2_ to the ozonation process enhances HO· formation and the radical reactions in the matrix [49]. Here, the positive correlation between percent degradation and H_2_O_2_ addition indicates a preferred indirect oxidation mechanism. 

Ozonation at basic pH also accelerates the formation of HO· because the presence of hydroxide ions can initiate ozone decomposition to form HO· [49]:O_3_ + HO^−^ → HO_2_^−^ + O_2_(3)
O_3_ + HO_2_^−^ → HO·+ O_2_·^−^ + O_2_

Therefore, the effect of ozonation with high-pH WWROC was expected to produce better TOrC degradation than at lower pH. Indeed, the result for ozonation at pH 10.5 demonstrated higher percent degradation for all of the TOrCs, under both ozonation intervals, than at pH 6.0, with and without H_2_O_2_ (with one exception—CBZ at the 5 min interval). These results highlight the selected TOrCs preference for the indirect mechanism.

Another important observation is the high efficiency of the indirect mechanism, where a lower ozone dose is required to achieve better degradation, in comparison to the direct mechanism. Figure 4 shows the dynamic ozone consumption by the samples, during 12 min ozonation for the three different conditions. The accumulated transferred ozone dose (TOD) during the first 2 min is about 50% higher for pH = 6, then pH = 10.5 and pH = 6 with H_2_O_2_, and the total slope from 2–12 min is 15% higher. This indicating that more ozone was required for potential oxidative reactions in the sample at pH = 6 only, while the two other conditions (addition of H_2_O_2_ and pH = 10.5) demonstrated lower ozone requirement, since the formed OH· also react with the compounds in the samples, lowering their total direct ozone oxidation potential. Calculated TOD/DOC ratios of ozonation with H_2_O_2_ (1.24 and 1.72 for the 5- and 12-min intervals, respectively) are smaller than the ratios of the other conditions (1.42 and 1.52 for 5 min; 1.72 and 2.23 for 12 min). Overall, this emphasizes the advantage of the ozone/H_2_O_2_ process over direct ozonation, as well over the high-pH adjustment of the matrix, since it is actually not practical to increase the WWROC pH to high values on a large scale.

The obtained results are consistent with other studies on wastewater and WWROC ozonation. Lakretz et al. [25] demonstrated, in their pilot system at SHAFDAN municipal WWTP (Israel), high degradation efficiency by O_3_/H_2_O_2_ (at a continuous flow rate, set to achieve a TOD/DOC ratio of 1.0–1.2) of pretreated secondary effluent, where the fast and moderate ozone-reacting compounds (same as in this study—SMX, CBZ, VLX, DCF, and BZF) were 96–100% degraded. The slow ozone-reacting compounds in their study (IHX as here, iopromide and iopamidol) were 41–81% degraded. However, in their case, the H_2_O_2_/DOC ratio was about 2.7, much higher than in the current study. Justo et al. [13] also applied ozonation on a one-stage WWROC matrix (with lower concentrations of TOC, COD, conductivity, and UVA_254_) at pH 8.3 (favoring indirect mechanisms), and demonstrated high degradation efficiency (~100%) for several fast-ozone-reacting TOrCs (including DCF, CBZ, and SMX) at a TOD/TOC value of 1.38. Degradation of the slow to moderate-reacting compound atenolol (K_O3_ ~ 1.7 × 10^3^ M^−1^ s^−1^ [50]) was about 80%. For lower ozone doses (TOD/TOC ratios of 0.82 and less), the percent degradation of the tested TOrCs decreased significantly (DCF ~65%, CBZ ~60%, and SMX ~80%). 

Overall, the trends from these two examples (and more) are similar to the trends in this study, demonstrating the efficiency of ozone (mainly by indirect mechanism) at degrading TOrCs in solid wastewater matrices.

An additional comparison between the untreated WWROC sample and O_3_/H_2_O_2_-treated WWROC is presented in Table 6. Several quality parameters were analyzed, and the results indicated the effectiveness of the O_3_/H_2_O_2_ treatment, mostly for reduction of organic parameters (color, UVA_254_ and DOC). However, this treatment was insufficient for inorganic pollutants, such as the toxic metals copper, manganese, and nickel, which were not removed at all, and the concentrations of which remained above the recommended and acceptable limits for reclaimed wastewater for irrigation in some countries, including Israel and the United States [51,52].

## 4. Conclusions

The efficiency of AOPs for the degradation of TOrCs in two-stage WWROC was demonstrated by comparing UV and ozone, with and without H_2_O_2_. Although WWROC concentrations in this work are several times higher than usual, we have demonstrated that O_3_ or UV with addition of H_2_O_2_ can be applied. Ozone or UV with addition of H_2_O_2_ demonstrated better oxidative conditions and breakdown of TOrCs due to formation of HO·, which is considerably more reactive than direct ozone or UV.

Even though O_3_/H_2_O_2_ was shown to be highly efficient for TOrC removal from the WWROC, and also improved other quality parameters indicating organic matter concentration, such as color, UVA_254_, COD, and TOC, this matrix was still rich in metals and nutrients. Some of the metal concentrations, found before and after ozone treatment, might be considered toxic to the environment. In addition, AOPs usually degrade some of the recalcitrant organic matter into more labile compounds, which are then available for additional biological degradation. Therefore, it is very important to continue investigating additional treatments for WWROC after the AOPs, in order to reduce metal ion concentrations, enable further biological treatment and achieve safe disposal.

## Figures and Tables

**Figure 1 materials-13-02785-f001:**
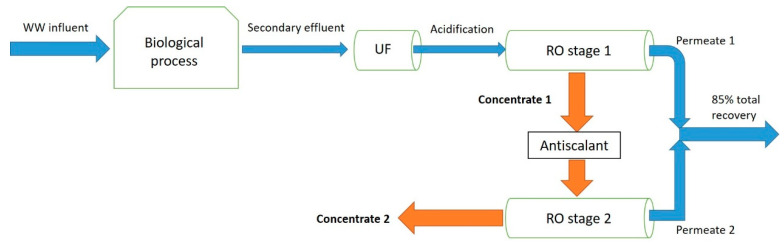
Schematic diagram of Bioreactor– ultrafiltration (UF)–reverse osmosis (RO)–RO Nir-Etzion wastewater treatment plant (WWTP) pilot system.

**Figure 2 materials-13-02785-f002:**
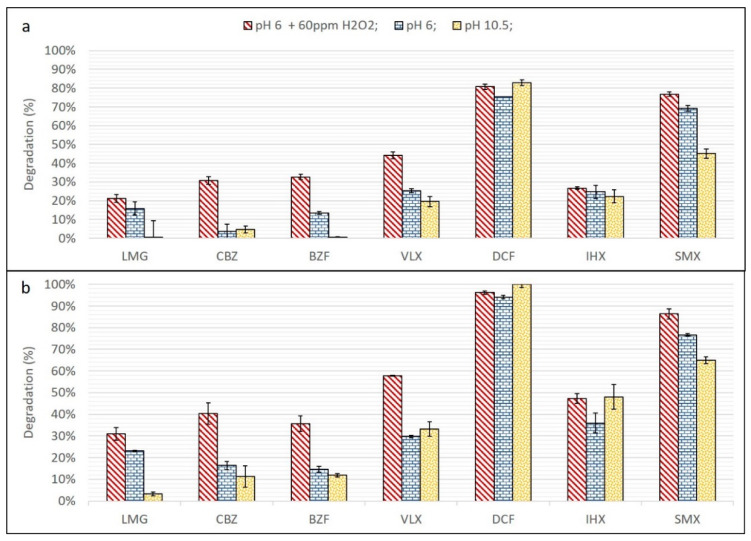
UV degradation for comparative experiments. (**a**) Fluence = 1730 mJ/cm^2^ and (**b**) fluence = 3460 mJ/cm^2^.

**Figure 3 materials-13-02785-f003:**
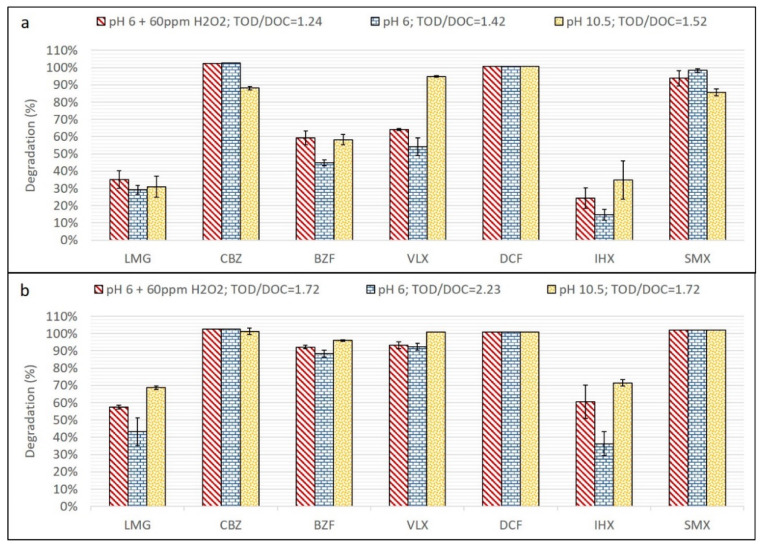
Results of degradation by ozone for comparative experiments: (**a**) 5 min ozonation and (**b**) 12 min ozonation.

**Figure 4 materials-13-02785-f004:**
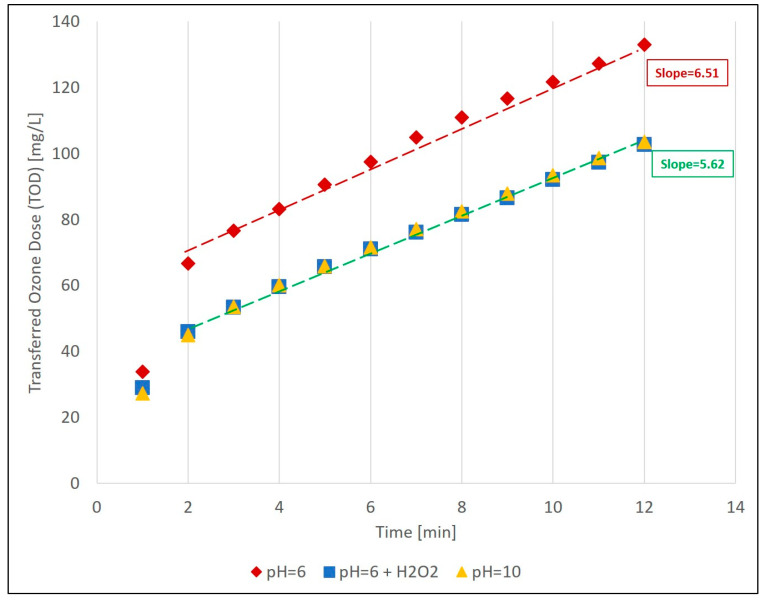
Dynamic progress of ozone consumption (as transferred ozone dose (TOD)) for the three studied conditions at 12 min interval. Slopes are calculated beween 2–12 min.

**Table 1 materials-13-02785-t001:** Selected trace organic compounds (TOrCs) in this study. The applied TOrCs concentrations are several times higher than were detected in secondary effluent.

Name	Class	K_O3_ (M^−1^s^−1^)	K_OH_ (10^9^ M^−1^s^−1^)	Conc. Found in Secondary Effluent, Shafdan, Israel (µg/L) [25]	Conc. in WWROC, Nir-Ezion, Israel (µg/L)
Iohexol (IHX) [25]	Contrast media	1.4	3.3	21.95–42.03	Not detected
Lamotrigine (LMG) [34]	Antiepileptic	4	2.1	Not tested	5.35 ± 0.05
Bezafibrate (BZF) [32]	Lipid regulator	590	7.4	0.10–0.15	1.41 ± 0.14
Venlafaxine (VLX) [35,36]	Antidepressant	3.3 × 10^4^	8.8	0.24–0.29	1.97 ± 0.05
Sulfamethoxazole (SMX) [32]	Antibiotic	2.5 × 10^6^	5.5	0.21–0.40	1.79 ± 0.25
Carbamazepine (CBZ) [32]	Antiepileptic	3 × 10^5^	8.8	0.87–1.04	11.26 ± 0.11
Diclofenac (DCF) [32]	Anti-inflammatory	1 × 10^6^	7.5	0.34–1.00	2.40 ± 0.20

**Table 2 materials-13-02785-t002:** Chemical parameters in a representative wastewater RO concentrate (WWROC) sample.

Parameter	Result
pH	5.9 + 0.1
Conductivity (mS/cm)	9.1 + 0.1
UVA_254_	1.49 + 0.1
DOC (mg/L)	63.8 ± 1.1
COD (mg/L)	183.5 ± 42.5
Magnesium (mg/L)	160.6 ± 2.5
Calcium (mg/L)	614.1 ± 4.0
Potassium (mg/L)	116.0 ± 7.3
Sodium (mg/L)	>1800
Iron (mg/L)	0.16 ± 0.001

**Table 3 materials-13-02785-t003:** Summary of applied methods for TOrCs degradation.

Treatment Method	WWROC Sample Conditions	Predictable Oxidation Mechanism
UV	pH = 6pH = 10.5 filtered 2.7 µm	Direct photolysis
UV/H_2_O_2_	pH = 6	By HO·(indirect)
O_3_	pH = 6pH = 10.5 filtered 2.7 µm	Direct ozone reactions for pH = 6; By HO (indirect) for pH = 10.5
O_3_/H_2_O_2_	pH = 6	By HO (indirect)

**Table 4 materials-13-02785-t004:** HPLC–MS chromatographic parameters of TOrCs.

Compound	IHX	LMG	SMX	VLX	CBZ	BZF	DCF
[M + H]	821.884	256.017	254.059	278.209	237.102	362.117	296.023
RT (min)	3.79	8.10	8.43	9.64	12.65	14.44	17.26

RT, retention time.

**Table 5 materials-13-02785-t005:** Instrumentation for quality measurements.

Instrument	Manufacturer	Model	Measurement
UV spectrophotometer	Varian	Cary 100	UV absorbance
pH m	Mettler Toledo	MA 235	pH
Conductivity m	IQ	IQ 170	Conductivity
Total organic carbon analyzer	O.I.	Aurora	DOC
Inductively coupled plasma	Spectro	Genesis	Metal ions
Photometer	Lovibond	MD 600	Chemical oxidation demand (COD), Color

**Table 6 materials-13-02785-t006:** Additional measurements before and after ozonation.

	WWROC	WWROC + O_3_ + H_2_O_2_
Visual	Dark yellow	Clear–pale yellow with light “cloudy” precipitation
Color (Pt-Co)	299	32
UVA_254_ (cm^−1^)	1.489	0.513
DOC (ppm)	63.8 ± 1.1	55.1 ± 0.8
COD (mgO_2_/L)	183.5 ± 42.5	101.5 ± 2.5
Copper (ppb)	540.7 ± 5.7	537.5 ± 4.9
Manganese (ppb)	223.8 ± 4.0	215.5 ± 4.0
Nickel (ppb)	650.2 ± 11.9	656.2 ± 10.0

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
