# Peer review of "Trace Organic Compound Removal from Wastewater Reverse-Osmosis Concentrate by Advanced Oxidation Processes with UV/O3/H2O2"

_materials, 2020, doi:10.3390/ma13122785_

Round 1
Reviewer 1 Report
In this manuscript, the authors investigated treatment of RO wastewater with advanced oxidation process. The paper is interesting and can be accepted by the current journal after minor revision.
However, the economic potentials of different AOPs should be compared and carefully discussed.
Author Response
Reviewer 1
In this manuscript, the authors investigated treatment of RO wastewater with advanced oxidation process. The paper is interesting and can be accepted by the current journal after minor revision. However, the economic potentials of different AOPs should be compared and carefully discussed.
Answer: Thanks for the suggestion. Indeed, the economic potential of these AOP’s technologies is an important issue, it was not an objective at the current study, thus, a comparison is not presented in the manuscript. This important suggestion may be evaluate in our further study and probably will be publish later on.

Reviewer 2 Report
This is a study about the investigation of AOP on the treatment of RO concentrate for pharmaceutical wastes till the realization of Zero Liquid Discharge. And the study showed some interesting to the readers with sound results, in view of practical engineering applications. Some major concerns should be addressed
1) How to clarify the direct oxidation and indirect oxidation mechanism in a complicated chemical reaction system in aqueous state? Especially how to clarify the functional portions of .OH and other radicals when O3 is applied?
2) The mechanism explanations should be explained with considering the chemical bonds for each chemical compounds, its chemical bond energy, chemical reaction in the liquid phase of solid/liquid interface?
3) How about the Opex for these AOP process? Like wise 1 ppm H2O2 for every ppm DOC? Then how about the O3 consumption for every ppm DOC. For engineering application, the O3 consumed/COD removed (or TOC removed) is very sensitive, how about this value in your study?
4) Performance comparison with other similar studies in this field should be provided.
5) What is the recommended post-AOP process to treat secondary effluent ?
6) Metal ions or other parameters in the aqueous plays a significant role in the activity of AOP, how about the impact of key ions (like Na+, Ca2+, Cl-, SO42-, etc) on the AOP process? And other parameters like temperature, hardness, conductivity?
7) Any other inhibition factors for AOP to be investigated here?
Author Response
Reviewer 2
This is a study about the investigation of AOP on the treatment of RO concentrate for pharmaceutical wastes till the realization of Zero Liquid Discharge. And the study showed some interesting to the readers with sound results, in view of practical engineering applications. Some major concerns should be addressed
1) How to clarify the direct oxidation and indirect oxidation mechanism in a complicated chemical reaction system in aqueous state? Especially how to clarify the functional portions of .OH and other radicals when O3 is applied?
Answer: Indeed, it is not an easy task to clarify between direct and indirect mechanisms. Sometimes the results you get is a sum of those oxidation mechanisms together. However, in order to evaluate each mechanism, we separate the treatments in that study, different pH and use a methods to evaluate the radical oxidation itself (pCBA). Additionally we differ the compounds by their KO3 and KOH to have various and different oxidation reactions, which we can follow upon.
2) The mechanism explanations should be explained with considering the chemical bonds for each chemical compounds, its chemical bond energy, chemical reaction in the liquid phase of solid/liquid interface?
Answer: Although we would be tremendously happy to include the suggested data and interpretations of the chemical oxidative mechanisms, this is not the scope of this work. Here, we would like to emphasize the possibility to use UV/H2O2 and ozone for TOrCs degradation in WWROC matrix and to suggest the preferable approach among them, in terms of degradation efficiency. We definitely want to research and publish such results in the near future.
3) How about the Opex for these AOP process? Like wise 1 ppm H2O2 for every ppm DOC? Then how about the O3 consumption for every ppm DOC. For engineering application, the O3 consumed/COD removed (or TOC removed) is very sensitive, how about this value in your study?
Answer: Calculating the Opex of such treatments is very complicate, depending on many parameters, some are direct to the process and some related to other industrial costs and varies between different setups and designs, the scale of the systems, local regulations etc. The scope of this work is only to demonstrate the possibility of AOP treatment for TOrCs degradation from the WWROC matrix. It emphasis the likelihood of AOP implementation on hard matrices and suggest the preferable AOP approach between the studied ones, in term of degradation efficiency.
Data regarding ozone consumption versus DOC was added to the ozone results section (figure 4) and DOC reduction is also appears on table 6.
4) Performance comparison with other similar studies in this field should be provided.
Answer: A comparison to results from similar studies present in literature appears on the results and discussion section, for both UV and ozone.
5) What is the recommended post-AOP process to treat secondary effluent ?
Answer: We believe that additional biological treatment after AOP is required for further elimination of the TOrCs degradation products formed during the oxidation process. This is mentioned on the conclusion section.
6) Metal ions or other parameters in the aqueous plays a significant role in the activity of AOP, how about the impact of key ions (like Na+,Ca2+, Cl-, SO42-, etc) on the AOP process? And other parameters like temperature, hardness, conductivity?
Answer: The scavenging rate of any matrix on AOPs is always an issue. We performed a study for the OH· scavenging rate by the WWROC, however presenting the methodology and calculations will burden this text and we think to publish this data as a different manuscript. Yet, we could not ignore from the reviewer justified comment and agreed that some reference should be given to this issue. Therefore, we added section "3.2 OH· scavengers" to discuss this issue, together with some supplementary data.
7) Any other inhibition factors for AOP to be investigated here?
Answer: In that study we chose to investigate the mentioned inhibition factors which we found relevant to our water quality. Those were mainly organics (DOC) and major ions present in the solution. In that study we didn’t have to cope (for example) with NO2-, which was not present in our investigated solution.

Reviewer 3 Report
Dear authors,
it appears that this paper is out of the topic as there are no traces of materials (and nanomaterials) as one average Materials journal reader would expect. However, after reading this paper more than a few times I was forced to revisit the Materials Journal focus. Therein, I have located the following: “Both academic and industrial views will be given for a better understanding of advanced oxidation processes and the fate of nanomaterials in water, to save our aquatic environments and protect human health, globally.” From this aspect, one could hardly agree that this paper fits the scope of this Materials special issue, however this paper indeed covers the “industrial view” topic.
Major issue
Line 65: This sentence is not clear as it follows that “photocatalysis using TiO2” is not AOP!? Please rephrase this sentence.
Lines 64-75: The authors clearly state that (pure) TiO2 is not suitable for TOrC removal. However, the authors should mention that the pure TiO2 is not efficient due to the recombination process. However, this recombination process is nowadays successfully hindered in TiO2-composites. So please comment on this. Having said that, can TiO2-composites be applied herein?
Lines 69-75: Please put these lines in the separate paragraph. Also, you should mention that, e.g. H2O2 is OH-radical scavenger and it blocks UV irradiation; and thus, you have decided to additionally study the UV/H2O2 -based AOPs… You can also refer to some ozone isuesses.
Lines 72-73: It appears that UV and O3 treatment of, e.g. CBZ, is more thoroughly studied herein: http://dx.doi.org/10.1080/09593330.2011.610359 . Please comment on this.
Line 115: Can you show C_03 (in-out) vs. time data and comment on the dynamic of the process? This would increase the scientific impact of your paper.
Line 163: CBZ was only marginally degraded by UV in this work. However, e.g., Dai et al. presented better CBZ degradation by UV (http://dx.doi.org/10.1080/09593330.2011.610359), especially when CBZ concentration was low. So, can poor CBZ (and BZF and VLX) degradation in this work be explained by the high CBZ (and BZF and VLX) concentration? Please clarify this in the text.
Lines 189-194: The equation in this paragraph has to be more visible (and numbered). This will increase overall scientific impact.
Line 232: Please put equations in separate lines.
Lines 269-272: It is expected that the addition of H2O2 will demonstrate better oxidative properties due to formation of OH-radical. So, you should also comment that a special attention should be focused on, e.g. concentration of H2O2 due to UV shielding and scavenging OH-radicals.
Section 2: Can you put all degradation methods used herein (and short descriptions) in one Table? This will facilitate the readers’ analyses.
Minor issues:
Line 21: Can you write like this? : However, AOPs are not sufficient to fully treat the WWROC; and thus, additional procedures are required to decrease metal ion and nutrient concentrations.
Line 48: Check for typos: NO3-
Line 50: Can you write like this? “Various methods are being studied and applied by both researchers and the industry to reduce the toxic elements in the WWROC or its final volume [17].”
Line 93: Table 2 is not clear. If I understand correctly, can you put “Conc. in WWROC (μg/L) “ values of “representative WWROC sample“ from Table 2 into an additional column in Table 1. Then, in Table 1 it is possible to compare “Concentrations found in secondary effluent in Israel ” values with ones used in this work?
Line 100: Typos; see “H2O2”
Line 101: Put DOC in parentheses.
Line 102: Please put here how pH was incresed.
Line 115: Please check units in Eq. (1). You have mg^-1 on the right side.
Line 115 : After Eq. (1) is coma (,) and after that “Where…” please clarify this.
Line 116: If necessary put units of C_O3 in parenthesis.
Overall, it seems that this paper has several issues. First, major and minor comments should be clarified. Second, the scientific level has to be boosted by introducing more arguments and discussions. The O3 part can be boosted by introducing (and commenting on) delta(O3) vs. time curves. Third, Section 2 is not organized well. So introducing Table with a list of methods will be beneficial for the reader. Fourth, the overall scientific impact of the paper will be increased if equations are better formatted (see comments). Fifth, you need to additionally stress out that your samples are obtained from a two-stage RO process; and thus, they are more concentrated (e.g. average [BZF] is 0.099-0.156 micro_g, whilst [BZG] in this work is 1.14 micro_g!?). This can be achieved by introducing your concentration data into Table 2. I would also suggest that the problem of higher concentrations is clearly stated also in Abstract. Sixth, in conclusion you should also comment that “although WWROC concentrations in this work was several times higher than usual, we have demonstrated that O3 or UV with addition of H2O2 can be applied….”
To summarize, if the authors resolve the aforementioned issues I am willing to review the paper once more.
Author Response
Reviewer 3
Dear authors,
it appears that this paper is out of the topic as there are no traces of materials (and nanomaterials) as one average Materials journal reader would expect. However, after reading this paper more than a few times I was forced to revisit the Materials Journal focus. Therein, I have located the following: “Both academic and industrial views will be given for a better understanding of advanced oxidation processes and the fate of nanomaterials in water, to save our aquatic environments and protect human health, globally.” From this aspect, one could hardly agree that this paper fits the scope of this Materials special issue, however this paper indeed covers the “industrial view” topic.
We aware that this is not an orthodox text for Materials, however we believe it could fit to the special issue "Advanced Oxidation Processes (AOPs) and Nanomaterials in Water Treatment and Purification".
Major issue
Line 65: This sentence is not clear as it follows that “photocatalysis using TiO2” is not AOP!? Please rephrase this sentence.
Answer: Accepted. We rephrased the sentence, emphasizing that TiO2 is AOP.
Lines 64-75: The authors clearly state that (pure) TiO2 is not suitable for TOrC removal. However, the authors should mention that the pure TiO2 is not efficient due to the recombination process. However, this recombination process is nowadays successfully hindered in TiO2-composites. So please comment on this. Having said that, can TiO2-composites be applied herein?
Answer: We admit that we are not specialized on TiO2 material, and thank the reviewer for the explanation. Therefore we did not intended for any statement regarding TiO2 as the reviewer mention. Some minor editorial changes were done in the text, so the effectiveness of TiO2 as AOP treatment for TOrCs removal is not ignored (a reference was also added to this sentence, line 65-69).
Lines 69-75: Please put these lines in the separate paragraph. Also, you should mention that, e.g. H2O2 is OH-radical scavenger and it blocks UV irradiation; and thus, you have decided to additionally study the UV/H2O2 -based AOPs… You can also refer to some ozone isuesses.
Answer: Lines putted in separate paragraph. We are mentioning the scavenging issues of H2O2 on results and discussion section.
Lines 72-73: It appears that UV and O3 treatment of, e.g. CBZ, is more thoroughly studied herein: http://dx.doi.org/10.1080/09593330.2011.610359 . Please comment on this.
Answer: Yes, we are aware that CBZ is well studied for different kinds of AOPs in different matrices for the past decades. However, as we mention on (lines 74-76…), the implementation of those AOPs on real WWROC is not been sufficiently studied. Dai et al. indeed perform a lot of study on CBZ, however they did not applied ozone at all and their study was done only in deionized water. Therefore it was difficult for us to compare and to refer their work in this manuscript.
Line 115: Can you show C_03 (in-out) vs. time data and comment on the dynamic of the process? This would increase the scientific impact of your paper.
Answer: Yes. We add figure 4 to show the data and discuss about the process and the different conditions in the O3/O3 H2O2 section.
Line 163: CBZ was only marginally degraded by UV in this work. However, e.g., Dai et al. presented better CBZ degradation by UV (http://dx.doi.org/10.1080/09593330.2011.610359), especially when CBZ concentration was low. So, can poor CBZ (and BZF and VLX) degradation in this work be explained by the high CBZ (and BZF and VLX) concentration? Please clarify this in the text.
Answer: As we mention previously, Dai et al. performed their study on a deionized water matrix which CBZ was the only organic compound and with very small amount of inorganics. For the best of our knowledge, this relatively clean matrix is probably the reason why the degradation of CBZ is better on their study.
Another reason for the superior CBZ degradation on Dai et al. work might be the different UV setup of the experiment. We accept that this reason should be discussed on our text and the manuscript is revised accordingly (line 228-231).
Regarding the suggestion of difference degradation rates due to high concentration, this conclusion is right if we examine Dai et al. work. However in our work the CBZ (and all other TrOCs) concentrations are much lower than on Dai et al. (we used 5-10µg/L concentrations, while they use minimum of 4.2µM CBZ, which is equivalent to 990µg/L) and the UV degradation is smaller. We think it is related to matrix issues rather than concentrations. For our opinion, this should not be discussed on the text, since the comparison to Dai et al. is not relevant due the different matrices and concentrations.
Lines 189-194: The equation in this paragraph has to be more visible (and numbered). This will increase overall scientific impact.
Answer: Done
Line 232: Please put equations in separate lines.
Answer: Done
Lines 269-272: It is expected that the addition of H2O2 will demonstrate better oxidative properties due to formation of OH-radical. So, you should also comment that a special attention should be focused on, e.g. concentration of H2O2 due to UV shielding and scavenging OH-radicals.
Answer: Scavenging of high H2O2 concentration on the UV photolysis is discussed on the UV and UV/H2O2 section. The reviewer refer here to the ozone section, where the H2O2 is not a scavenger (no UV), therefore the requested comment is not applicable here.
Section 2: Can you put all degradation methods used herein (and short descriptions) in one Table? This will facilitate the readers’ analyses.
Answer: We accept and add the proposed table (table 3) on this section, along with some explanations (Lines 118-121…)
Minor issues:
Line 21: Can you write like this? : However, AOPs are not sufficient to fully treat the WWROC; and thus, additional procedures are required to decrease metal ion and nutrient concentrations.
Answer: test revised accordingly.
Line 48: Check for typos: NO3-
Answer: Done
Line 50: Can you write like this? “Various methods are being studied and applied by both researchers and the industry to reduce the toxic elements in the WWROC or its final volume [17].”
Answer: Text revised accordingly
Line 93: Table 2 is not clear. If I understand correctly, can you put “Conc. in WWROC (μg/L) “ values of “representative WWROC sample“ from Table 2 into an additional column in Table 1. Then, in Table 1 it is possible to compare “Concentrations found in secondary effluent in Israel ” values with ones used in this work?
Answer: Done
Line 100: Typos; see “H2O2”
Answer: Done
Line 101: Put DOC in parentheses.
Answer: Done
Line 102: Please put here how pH was incresed.
Answer: 1N NaOH. Text revised accordingly.
Line 115: Please check units in Eq. (1). You have mg^-1 on the right side.
Answer: Equation units corrected
Line 115 : After Eq. (1) is coma (,) and after that “Where…” please clarify this.
Answer: It was a typo. Text corrected
Line 116: If necessary put units of C_O3 in parenthesis.
Answer: No need. Corrected in the equation
Overall, it seems that this paper has several issues. First, major and minor comments should be clarified. Second, the scientific level has to be boosted by introducing more arguments and discussions. The O3 part can be boosted by introducing (and commenting on) delta(O3) vs. time curves. Third, Section 2 is not organized well. So introducing Table with a list of methods will be beneficial for the reader. Fourth, the overall scientific impact of the paper will be increased if equations are better formatted (see comments). Fifth, you need to additionally stress out that your samples are obtained from a two-stage RO process; and thus, they are more concentrated (e.g. average [BZF] is 0.099-0.156 micro_g, whilst [BZG] in this work is 1.14 micro_g!?). This can be achieved by introducing your concentration data into Table 2. I would also suggest that the problem of higher concentrations is clearly stated also in Abstract. Sixth, in conclusion you should also comment that “although WWROC concentrations in this work was several times higher than usual, we have demonstrated that O3 or UV with addition of H2O2 can be applied….”
Answer: all the suggested issues have been addressed
To summarize, if the authors resolve the aforementioned issues I am willing to review the paper once more.

Round 2
Reviewer 2 Report
The revised manuscript at present state is ready for publication.
Author Response
The revised manuscript at present state is ready for publication.
Thanks!!
Reviewer 3 Report
The soundness of the paper is now increased. Now, it is obvious that this paper deals with higher concentrations than usual.
There are still minor typing errors and Table quality errors that should be resolved prior to publication.
Please find some minor issues that should be clarified:
Line 82: Table 1. Selected TOrCs in this study. The applied TOrCs concentration is several times higher than…..
Line 105 …to result in significant…?
Line 135: Table 3 should be smoothed to meet publication quality level.
Line 308: Table 6 should be smoothed to meet publication quality level.
Line 312: See typos H2O2
Overall, the paper is improved and if the authors clarify the aforementioned issues there is no need for an additional review.
Author Response
Reviewer 3
Thanks for the comments
Line 82: Table 1. Selected TOrCs in this study. The applied TOrCs concentrations are several times higher than
Answer: Done
Line 105 …to result in significant…?
Answer: Done
Line 135: Table 3 should be smoothed to meet publication quality level.
Answer: Done
Line 308: Table 6 should be smoothed to meet publication quality level.
Answer: Done
Line 312: See typos H2O2
Answer: Done
